# McArdle Disease: Clinical, Biochemical, Histological and Molecular Genetic Analysis of 60 Patients

**DOI:** 10.3390/biomedicines8020033

**Published:** 2020-02-15

**Authors:** Pushpa Raj Joshi, Marcus Deschauer, Stephan Zierz

**Affiliations:** 1Department of Neurology, Martin-Luther-University Halle-Wittenberg, 06120 Halle (Saale), Germany; marcus.deschauer@mri.tum.de (M.D.); stephan.zierz@uk-halle.de (S.Z.); 2Department of Neurology, School of Medicine, Technical University Munich, 81675 Munich, Germany

**Keywords:** McArdle, second-wind, myophosphorylase, mutation, permanent weakness

## Abstract

A clinical, biochemical, histological and molecular genetic analysis of 60 McArdle patients (33 males and 27 females; mean age at diagnosis: 37 years) was performed. The objective of this study was to identify a possible genotype–phenotype correlation in McArdle disease. All patients complained of exercise-induced myalgia and fatigue; permanent weakness was present in 47% of the patients. Five percent of patients conveyed of masticatory muscle weakness. Age of onset was <15 years in 92% patients. Serum creatine kinase was elevated 5 to13-fold. Forearm ischemic test showed decreased lactate production but excessively increased ammonia upon exercise (*n* = 16). Muscle biopsies revealed highly reduced or missing myophosphorylase activity (*n* = 20) (mean: 0.17 ± 0.35 U/g tissue; normal: 12–61) and histologically, sub-sarcolemmal glycogen accumulation (*n* = 9). Molecular genetic analysis revealed the common p.Arg50Ter mutation in 68% of the patients. Other rather frequent mutations were p.Arg270Ter (allele frequency: 5%) followed by c.2262delA and p.Met1Val (allele frequencies: 3%). Twenty-four other rare mutations were also identified. No genotype–phenotype correlation was observed. The analysis highlights that testing of the p.Arg50Ter mutation could be performed first in molecular genetic testing of patients with exercise intolerance possibly due to McArdle disease. However, there is enormous mutation heterogeneity in McArdle disease thus sequencing of the myophosphorylase gene is needed in patients highly suspicious of McArdle disease.

## 1. Introduction

McArdle disease (Glycogen storage disease type V; GSD5; MIM #232600) is one of the most common metabolic myopathies that is caused due to lack of the enzyme myophosphorylase (skeletal muscle isoform of glycogen phosphorylase). Clinically, the disease is characterized by exercise-induced myalgia, cramps, myoglobinuria and also permanent weakness after longer disease duration [1,2,3,4]. Molecular genetically, a common truncating mutation p.Arg50Ter (previously denoted as p.Arg49Ter) in the myophosphorylase gene (*PYGM*) is associated with more than 50% of the mutant alleles in Caucasian patients [1,2,3,4,5,6,7]. Additionally, more than 150 rare mutations have been reported according to the Human Gene Mutation Database [8]. Mode of inheritance of McArdle disease is autosomal recessive. A clear genotype–phenotype correlation based on various mutations in McArdle disease has not been established [2,3,7,9].

Despite being one of the most frequent glycogen storage diseases [10], there are only few large cohort studies (>50 patients) on McArdle diseases [3,11,12,13]. In these studies, which were conducted with patients mainly from Spanish and French origin, molecular genetic and clinical features of the patients are elaborately described. However, the biochemical and histological aspects are only marginally mentioned. In the present study, clinical, biochemical, histological and molecular genetic analysis of 60 McArdle patients (47 German patients) was performed to analyse whether the clinical and biochemical phenotypes are influenced by the underlying genotypes.

## 2. Material and Methods

### 2.1. Patients

Clinical, biochemical and molecular genetic data of 60 patients with McArdle disease were analysed. Clinical and molecular genetic data of 56 patients are previously described [2]. However, forearm lactate ischemic test, biochemical and histological data were not included in that study. All patients were of Caucasian origin with majority of patients being Germans (*n* = 47). The other 13 patients were from UK (*n* = 5), Norway (*n* = 2), Turkey (*n* = 2), Italy (*n* = 1), Austria (*n* = 1), Lebanon (*n* = 1) and Syria (*n* = 1). Two patients had consanguinity in the family. Thirty-three patients were males and 27 patients were females. The mean age at the time of diagnosis was 37 years (range: 13 to 76 years).

The study was performed retrospectively by gathering the data of patients that were diagnosed in our department since 1994. All patients and the legal guardian of minors have given written consent that their results can be anonymously published for scientific and academic purposes. The study has been performed in accordance with the ethical standards laid down in the 1964 Helsinki declaration.

### 2.2. Clinical Characterisation

The clinical data of patients were collected retrospectively either by personally interviewing the patients in our department or through clinical reports of referring physicians. During the interview, the patients were asked the question based on their diseased status such as age of onset, family history, presence/absence of permanent weakness, presence of other symptoms (exercise intolerance, myalgia and fatigue) and ability to resume exercise after a short period of reduced work load intensity (second-wind phenomenon).

### 2.3. Laboratory Studies

Serum creatine kinase (CK) levels were extracted from the archived clinical files of patients (*n* = 33). In addition, levels of lactate and ammonia upon exercise were measured in 16 available patients according to the established forearm ischemic exercise test protocol [14,15]. The results were compared with those of 21 healthy controls (14 males, 7 females; mean age: 38 years).

### 2.4. Biochemical and Histological Analysis of Muscle Biopsies

Muscle tissues of 20 patients were obtained in open biopsy in our department and stored in liquid nitrogen for further analysis. The activity of myophosphorylase in muscle homogenate of patients and healthy controls (*n* = 20, 11 males, 9 females; mean age: 38 years) was then photometrically accessed following the modified method described by Bergmeyer [16].

In 9 available biopsies, Periodic acid–Schiff (PAS) staining was performed to analyze glycogen content by sequentially incubating the sections with Periodic acid and Schiff’s reagents following previously described staining protocol [17]. A histochemical analysis of myophospharylase activity in muscle biopsy was done by performing myophosphorylase staining using previously described method [18]. The results were compared with those of healthy controls (*n* = 20, 11 males, 9 females; mean age: 38 years).

### 2.5. Molecular Genetic Analysis

Genomic DNA was extracted from the muscle or blood of all patients using standard protocol. The p.Arg50Ter mutation was screened in all patients using previously described restriction fragment length (RFLP) method [5]. In patients who were heterozygous or were negative for this frequent mutation, coding regions of the *PYGM* gene including exon–intron boundaries were sequenced [19]. Sequences were compared with the genomic structure of the *PYGM* gene [17].

### 2.6. Genotype-Phenotype Correlation

Genotype–phenotype correlation was analyzed between two groups: (i) patients with homozygous p.Arg50Ter mutation (*n* = 27) and patients with heterozygous p.Arg50Ter mutation (*n* = 14) and (ii) patients with p.Arg50Ter mutation on at least one allele (*n* = 41) and patients without p.Arg50Ter mutation on both alleles (*n* = 19). A chi-square test was performed to compare the groups.

## 3. Results

### 3.1. Clinical Features

All patients complained of exercise-induced myalgia and almost half of the patients (47%) suffered from permanent weakness. Five percent patients suffered from masticatory muscle weakness and 7% patients had cramps (Table 1).

Despite early onset (<15 years) in majority of patients (87%), the mean age at diagnosis was 37 (±14) years in our cohort. Fifteen patients (25%) conveyed of positive family history of McArdle disease. ‘Second-wind’ phenomenon was reported in more than half of patients (56%). Table 2 illustrates detailed clinical features in our cohort that are compared with clinical features of McArdle patients in previously published reports [20,21].

### 3.2. Laboratory Tests

Serum creatine kinase was elevated 5 to 18-fold the upper reference range limit in all available patients (*n* = 33). In forearm lactate ischemic test, the delta lactate in sixteen tested patients (mean: 0.5 mmol/L) was significantly lower (*p* < 0.001) than that of controls (mean: 3.60 mmol/L). On the other hand, delta ammonia was almost seven times more in patients (mean: 300.9 μmol/L) than in controls (mean: 45.5 μmol/L) with a significance level of *p* < 0.001 (Figure 1).

### 3.3. Biochemical Features

The activity of enzyme myophosphorylase was extremely reduced in muscle homogenates of all 20 patients whose muscle biopsies were available for biochemical analysis (mean: 0.17 ± 0.35 U/g tissue; median: 0.0; normal: 12–61). In 12 patients, there was complete loss of enzyme activity.

### 3.4. Histological Findings

The histological investigation showed marked sub-sarcolemmal glycogen deposition in Periodic acid-Schiff (PAS) staining in all tested muscle biopsies (*n* = 9) (Figure 2A). Similarly, hiostochemical staining of enzyme myophosphorylase showed no or minimal enzyme activity in all tested biopsies in comparison to healthy controls (Figure 2B,C).

### 3.5. Molecular Genetic Features

Molecular genetic analysis identified the p.Arg50Ter mutation in 41 patients (68%) in at least one allele with an allele frequency of 0.57. Forty-five percent patients were homozygous for this common mutation. Other rather frequent mutations were p.Arg270Ter (allele frequency: 5%) followed by c.2262delA, p.Met1Val (allele frequencies: 3%) and Gly686Arg (allele frequency: 2%). Apart from these 5 mutations, 23 other rare mutations were also identified: 2 mutations (Arg94Trp and Glu349Lys) each in two unrelated patients and 21 mutations in single cases. A detailed genetic overview of the patients is illustrated in Table 3.

### 3.6. Genotype–Phenotype Correlation

A significant statistical correlation was not identified in the frequency of permanent weakness patients with p.Arg50Ter mutation on both alleles (14/27 patients) and the p.Arg50Ter mutation on only one allele (5/14 patients) as well as patients with p.Arg50Ter mutation on at least one allele (19/41 patients) and patients with other than the p.Arg50Ter mutation on both alleles (11/19 patients). In both these genotype groups, significant differences were not identified in terms of age of onset, laboratoty tests and biochemical and histological outcomes.

## 4. Discussion

### 4.1. Clinical Features

Exercise intolerance has been unanimously identified as the classical symptom associated with McArdle disease. This has been confirmed by the fact that all our patients complained of exercised-induced myalgia and fatigue. In the present cohort, 56% patients conveyed that they were able to resume exercise after a short period of reduced work-load intensity at the first appearance of fatigue. This emphasizes that ‘second-wind’ phenomenon is frequently observed in McArdle patients [3,19]. 

A striking 47% patients in our cohort reported permanent weakness in comparison to only 28% in the meta-analysis of other previous studies [20,21] (Table 2). All eight patients who had late onset (three patients: later than 50 years) had permanent weakness. This is consistent with other findings with permanent muscle weakness identified frequently after especially 40 years [17,18,19]. In all patients with permanent weakness, this was localized in limbs (entire arm and leg regions, shoulders, hands, thighs, calves). Four patients (7%) complained of cramps (17%) (Table 1). Cramps have also been previously reported in 87% McArdle patients [10,22,23]. Three patients conveyed masticatory muscle weakness especially during chewing (5%). These findings show that cramps and masticatory muscle weakness are not that rare in metabolic myopathies as we had recently reported these clinical features also in patients with CPT II deficiency [24]. 

The onset of the disease was seen more frequently in early life (<15 years), although five late-onset patients were also identified. This is in line with other reports on McArdle diseases [20,21]. Despite the early emergence of muscle symptoms in our patients, the mean age at diagnosis was 37 years (range 13–76 years). After emergence of the first symptoms (mean: 9 years, range: 1–63 years), it took an average of 28 years (range: 0–71 years) to have a diagnosis of McArdle disease. Hence, it can be assumed that a lot of McArdle patients are underdiagnosed or misdiagnosed. This point is strengthened by the fact that most of the McArdle patients do not have evidence of severe muscular involvement such as atrophies or other abnormalities. Hence, at least seven of our patients conveyed that their disabilities had initially been misdiagnosed as psychosomatic disorders. This reflects the diagnostic impediment associated with neuromuscular disorders [24]. McArdle disease is one of the most common metabolic myopathies with an estimated prevalence of ~1/100,000 in European population. However, a recent study based on next-generation sequencing of PYGM in subjects with no primary muscle disease suggested that this prevalence is an underestimate and the actual prevalence would be more than twice as high in this population [10].

The mode of inheritance of McArdle disease is autosomal-recessive. However, rarely pseudo-dominantly inherited patients are also reported [25,26]. Only 26% patients in our study conveyed of positive family history in contrast to 53% in meta-analysis of other studies that contained mainly patients from USA, Spain and France (Table 1). An explanation for this discrepancy could be that 78% patients in our cohort were Germans and generally, German families are comparatively smaller than families in other countries like Spain and France (https://www.oecd.org).

### 4.2. Laboratory Findings

Usually, a CK elevation above 4-fold the upper limit of normal range is considered to be pathogenic in muscle disorders [27]. All the patients in our study had elevated levels of CK clearly above 5-fold the normal range, showing that the elevated levels of CK should be considered as one of the preliminary diagnostic criteria in diagnosis of McArdle disease, if the patients have clear clinical symptoms that are in line with McArdle disease.

In line with our forearm ischemic test outcome, previous studies on McArdle patients also reported an excessive increase of ammonia upon physical exercise in serum [28,29,30,31]. Recently, we had postulated that the increased ammonia production upon exercise in McArdle patients is not due to enzyme induction but could be due to kinetic regulation of the enzyme AMP deaminase (AMPD) which is based on hyperbolic response of enzyme activity to substrate concentration as described by the Michaelis Menten kinetics [32]. Lactate forearm ischemic test is one of the efficient diagnostic tools in different myopathies [15] and hence lacking lactate increase and excessively increased ammonia in patients with glycogenosis gives a clear hint of McArdle diseases.

### 4.3. Biochemical Features

Irrespectively of the underlying mutations, extremely reduced myophosphorylase activity was detected in muscle homogenates of patients. It has been assumed that in McArdle disease, the mutated alleles are either not transcribed or not translated to protein [33]. This confirms that unlike in other metabolic myopathies such as CPT II deficiency (that occurs due to abnormal regulation of the enzyme) [23], McArdle disease presents with almost complete lack of enzyme myophosphorylase. However, rare symptomatic heterozygote patients with intermediate myophosphorylase have also been sporadically reported [25].

### 4.4. Myohistological and Histochemical Features

Sub-sarcolemmal glycogen accumulation has been established as the myohistological diagnostic feature and reduced or no activity of enzyme myophosphorylase in muscle biopsy as histochemical diagnostic feature in McArdle disease [9,18,21]. Myohistological and histochemical findings revealing subsarcolemmal glycogen accumulation and lack of myophosphorylase activity in muscle biopsies of our patients further strengthen these notions.

### 4.5. Molecular Genetic Features

A total of 28 different mutations spreading throughout the *PYGM* gene were identified, although the ‘common’ p.Arg50Ter mutation was identified in at least one allele in more than two-third patients (Table 3). Our results are in line with previous findings as this mutation was reported frequently (allele frequency: 0.31–0.81) in patients of European origin [5,6,7,11,34]. The patients reported in these studies are mainly of Spanish, French, Dutch, UK and Italian origin. Apart from European patients, this mutation has also been identified in about 60% US American [5,34] and 50% Brazilian patients [35]. This mutation has never been identified in Japanese patients [36]. In our cohort, this mutation was also not identified in two non-European patients (Syrian: p.Ser450Leu homozygous and Lebanese: p.Met1Val homozygous). Neither was this mutation identified in our two Turkish patients. This strengthens the strong ‘ethnic’ European origin of the p.Arg50Ter mutation considering the European ancestry of US American and Brazilian patients. Five other mutations were reported in more than one family. Three mutations were present in single cases but were also identified in other studies. This shows that most of the McArdle mutations identified in our cohort are not private mutations.

### 4.6. Genotype–Phenotype Correlation

Higher peak workload and oxidative capacity in patients with the p.Arg.50Ter and p.Gly205Ser mutations compared to patient with splice-site mutations (c.425-26A>G and c.856-601G>A) were reported, suggesting a possible genotype–phenotype correlation [37]. In line with this report only one patient with homozygous splice-site mutation (c.772-2A>T) in our cohort also had severe permanent shoulder weakness. These findings hint towards the severity of homozygous splice-site mutations over other mutations as the other two patients with heterozygous splice-site mutation (both compound heterozygous for p.Arg50Ter and c.1239+1G>A) had milder symptoms. However, large cohort studies including broad spectrum of *PYGM* mutations show the lack of distinct genotype-phenotype correlation [2,3,12,38]. In the present study, a clear genotype–phenotype correlation could not be established either as so many different genotypes were observed and all our patients presented with typical signs of exercise intolerance. Patients with permanent weakness also had broad mutation spectrum. In addition, genotype–phenotype correlations based on laboratory tests, biochemical and histological outcomes were also not identified.

## 5. Conclusions

McArdle disease is one of the most common glycogen storage disease that presents with exercise intolerance and less frequently permanent weakness. The first symptoms are experienced almost exclusively in childhood or early adulthood. However, rare late onset patients presenting the symptoms as late as in sixth decade are also known. The peculiar laboratory test feature associated with McArdle diseases is no or minimal production of lactate but excessive production of ammonia upon exercise. McArdle patients have almost total lack of enzyme myophosphorylase in skeletal muscle. A clear genotype–phenotype correlation is not seen as McArdle disease has an enormous mutation spectrum. A p.Arg50Ter mutation is; however, frequent in European patients. Hence, the screening of this mutation has been suggested as the first line of diagnosis in European patients, especially in patients with exercise intolerance. If the patients have clear suspicion of McArdle disease, sequencing of the whole *PYGM* gene is recommended.

## Figures and Tables

**Figure 1 biomedicines-08-00033-f001:**
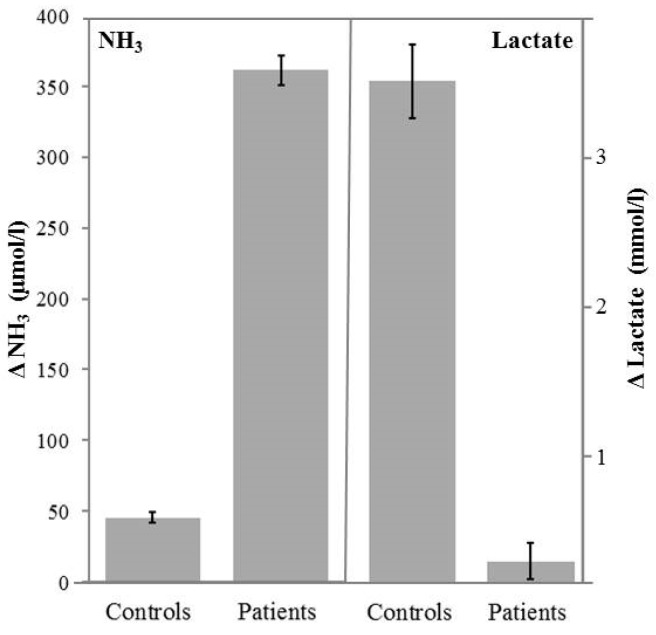
Forearm lactate ischemic test showing significantly higher (*p* < 0.001) ammonia production but significantly lower (*p* < 0.001) lactate production upon exercise in patients with McArdle disease (*n* = 16) compared to healthy controls (*n* = 21). The bar charts represent mean values of delta ammonia and delta lactate in patients and healthy controls respectively. The error bars represent standard errors.

**Figure 2 biomedicines-08-00033-f002:**
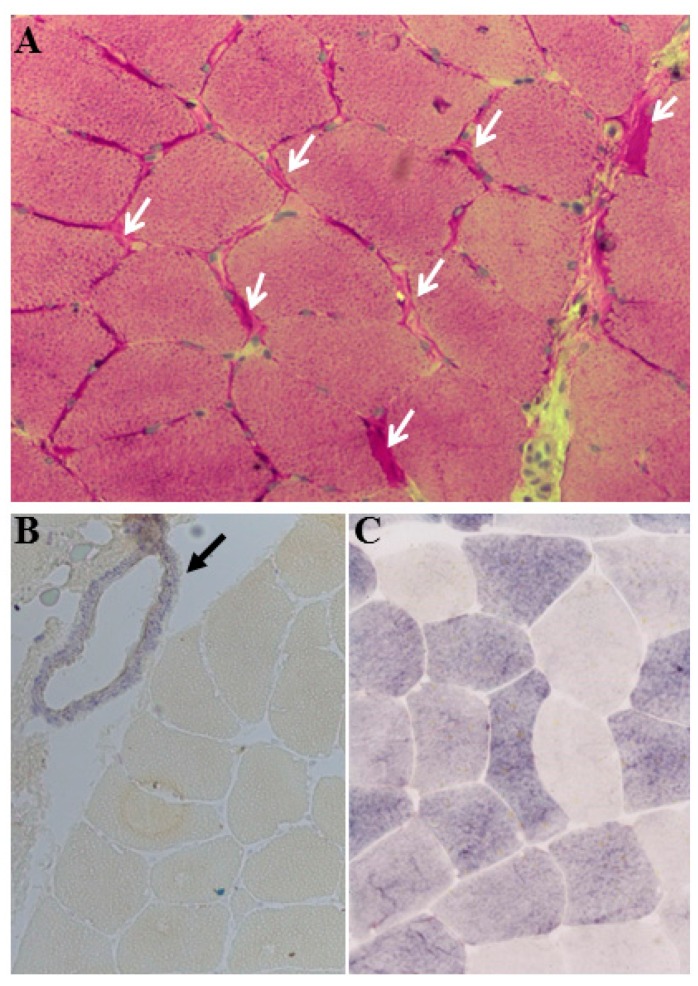
Myohistological and histochemical features of the muscle biopsy (200× magnification) of a patient with p.Arg50Ter mutation in homozygous state showing (**A**) subsarcolemmal glycogen accumulation (white arrows), (**B**) complete loss of myophosphorylase activity in patient’s muscle biopsy (although phosphorylase on the smooth muscle cells of blood vessel is stained; shown by an arrow) in comparison to (**C**) positive myophosphorylase reaction in muscle biopsy of healthy control.

**Table 1 biomedicines-08-00033-t001:** Genotype and clinical data of patients with permanent weakness.

Patient (Gender)	Genotype	Onset	Family History	Permanent Weakness	Cramps	Masticatory Muscle Involvement
1 (M)	p.Arg50Ter/p.Arg50Ter	Childhood	−	Thighs	−	−
2 (M)	p.Arg50Ter/p.Arg50Ter	Childhood	−	Arms, legs	**+**	−
3 (F)	p.Arg50Ter/p.Arg50Ter	Childhood	−	Thighs	−	−
4 (M)	p.Arg50Ter/p.Arg50Ter	Childhood	Brother	Thighs, Arms	−	−
5 (F)	p.Arg50Ter/p.Arg50Ter	Childhood	−	Legs	−	−
6 (M)	p.Arg50Ter/p.Arg50Ter	Childhood	−	Arms, legs	−	−
7 (M)	p.Arg50Ter/p.Arg50Ter	Adolescence	−	Arms, legs	−	−
8 (M)	p.Arg50Ter/p.Arg50Ter	Late (50 Yrs.)	Cousin	Shoulder, arm	−	−
9 (M)	p.Arg50Ter/p.Arg50Ter	Late (50 Yrs.)	−	Arms, legs	−	−
10 (M)	p.Arg50Ter/p.Arg50Ter	Childhood	−	n.a.	−	−
11 (F)	p.Arg50Ter/p.Arg50Ter	Childhood	Father	n.a.	−	−
12 (F)	p.Arg50Ter/p.Arg50Ter	Childhood	−	n.a.	−	−
13 (F)	p.Arg50Ter/p.Arg50Ter	Childhood	−	n.a.	−	−
14 (F)	p.Arg50Ter/p.Arg50Ter	Childhood	−	n.a.	−	−
15 (M)	p.Arg50Ter/p.Asp129His	Childhood	−	Thighs, calves	−	−
16 (F)	p.Arg50Ter/p.Arg139Trp	Childhood	−	Arms, legs,	−	**+**
17 (M)	p.Arg50Ter/p.Glu655Lys	Childhood	−	Arms, legs, back	−	−
18 (F)	p.Arg50Ter/p.Gly686Arg	Childhood	Sister	Arms, legs	−	−
19 (F)	p.Arg50Ter/p.Gly686Arg	Youth	−	Arms, legs	−	−
20 (M)	p.Met1Val/p.Met1Val	Adolescence	Several	Shoulders	−	−
21 (F)	p.Met1Val/p.Met1Val	Youth	−	Arms, legs	−	−
22 (M)	c.25delG/c.25delG	Youth	Several	Arms, legs	−	−
23 (F)	p.Tyr85Ter/p.Arg94Trp	Childhood	−	Arms, legs	**+**	**+**
24 (F)	p.Arg94Trp/p.Arg94Trp	Late (63 Yrs.)	−	Thighs, Toe	**+**	−
25 (F)	p.Arg270Ter/p.Arg270Ter	Childhood	Brother	Arms, legs	**+**	−
26 (M)	IVS6-2A>T/IVS6-2A>T	Childhood	Sister	Shoulders	−	−
27 (F)	p.Gln337Arg/p.Gln337Arg	Childhood	−	Arms, legs, back	−	−
28 (F)	p.Tyr574Ter/c.1948delC	Childhood	−	Arms, legs,	−	**+**
29 (M)	p.Met680Val/c.2262delA	Childhood	Several	Shoulders	−	−

M: male, F: female, +: presence of cramps and/or masticatory muscle involvement, -: negative family history or absence of cramps and/or masticatory muscle involvement, Several: Several family members affected, n.a.: not available, Childhood: 1–12 years, Adolescence: 13–18 years, Youth: >19 years.

**Table 2 biomedicines-08-00033-t002:** Clinical features of McArdle patients in present cohort compared to meta-analysis of previous studies.

Clinical features	Present Study (*n* = 60)	Other Studies [20,21] (*n* = 166)	
	*n* (%)	*n* (%)	*p*
Exercise intolerance	60 (100)	146/166 (89)	n.s.
Permanent weakness	28 (47)	46/166 (28)	**0.02**
Age at onset (Years)			
<15	52 (87)	94/112 (85)	n.s.
15–30	5 (8)	10/112 (9)	n.s.
>30	3 (5)	6/112 (6)	n.s
Age at diagnosis (Years)			
<10	0 (0)	5/112 (4)	n.s.
10–30	19 (32)	55/112 (50)	**0.028**
30–50	32 (53)	30/112 (27)	**<0.001**
>50	9 (15)	21/112 (19)	n.s.
Positive family history	15 (25)	51/112 (53)	**0.026**
CK elevation	33/33 (100)	54/54 (100)	n.s.
Second wind	19/34 (56)	n.a.	-

*p*: two tailed probability coefficient (chi-square test) with level of significance set to ≤ 0.05, n.a.: data not available, statistically significant p value are marked by bold cases, n.s.: not significant, mean age of onset in our study: 9 (±14) years, mean age at diagnosis in our study 37 (±14) years.

**Table 3 biomedicines-08-00033-t003:** Genotype and ethnic origin characterization of patients (splice-site mutations are marked by asterisk).

No. of Patients	Allele 1	Allele 2	Ethnic Origin
27	p.Arg50Ter	p.Arg50Ter	Germany (*n* = 23), UK (*n* = 4)
4	p.Arg50Ter	p.Arg270Ter	Germany (*n* = 4)
2	p.Arg50Ter	p.Gly486Asp	Both Norway
2	p.Arg50Ter	p.Gly686Arg	Both Germany
2	p.Arg50Ter	c.1239+1G>A*	Both Germany
2	p.Met1Val	p.Met1Val	Turkey (*n* = 1), Lebanon (*n* = 1)
1	p.Arg50Ter	p.Asp129His	Germany
1	p.Arg50Ter	p.Arg139Trp	Germany
1	p.Arg50Ter	p.Glu655Lys	Germany
1	p.Arg50Ter	c2262delA	Germany
1	c.25delG	c.25delG	Germany
1	c.78_79delTG	c.78_79delTG	Germany
1	p.Tyr85Ter	p.Arg94Trp	Germany
1	p.Arg94Trp	p.Arg94Trp	UK
1	p.Gly157Val	p.Glu349Lys	Germany
1	p.Gly205Ser	p. Arg576Ter	Germany
1	p.Arg270Ter	p.Glu349Lys	Germany
1	p.Arg270Ter	p.Arg270Ter	Italy
1	c.772-2A>T*	c.772-2A>T*	Germany
1	p.Gln337Arg	p.Gln337Arg	Germany
1	p.Glu384Lys	c.1155_1156delGG	Germany
1	p.Ser450Leu	p.Ser450Leu	Syria
1	p.Leu397Pro	c2262delA	Germany
1	p.Arg570Trp	p.Gly686Arg	Austria
1	p.Tyr574Ter	c.1948delC	Germany
1	p.Gln666Glu	c.2262delA	Germany
1	p.Met680Val	c2262delA	Turkey

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
