# Peer review of "McArdle Disease: Clinical, Biochemical, Histological and Molecular Genetic Analysis of 60 Patients"

_biomedicines, 2020, doi:10.3390/biomedicines8020033_

Round 1

Reviewer 1 Report

The paper by Joshi et al presents data fra om a cohort of 60 patients with McArdles disesae (glycogen storage disease type V/myophosphorylase deficiency). The paper is well written and clearly presents data.

However, there are some concerns about the paper. This present paper is based mainly on patients, whose genotype and clinical features were presented in 2007 by Deschauer et al. The new data is mainly histology and biochemistry and a revisit of the muscle weakness, also for a few added patients. This means that part of the data has already been published.

The presented histology and biochemical data does not contribute to the body of knowledge, since it has been known for a long time that in patients with McArdle disease lactate is low, ammonia is high due to alternate ATP production and periodic acid Schiff stain will demonstrate patches of glycogen in muscle sections.

In the original 2007 paper 15 of 40 patients had permanent weakness (37.5%), the present report states that 47% have permanent weakness. The meta-analysis demonstrates a 28% permanent weakness and a report from 2009 by Nadaj-Pakleza including 80 patients of which 11% had muscle weakness (rising to 37.5% if only counted above the age of 40). At this point it is generally accepted that there is no genotype-phenotype relationship, the 2007 paper states this as well. The muscle weakness generally affects arms and legs and it is possible that permanent weakness is related to age. 

It is of interest to the clinician how muscle weakness develops over time. However, it is my understanding that the presented clinical data (muscle weakness) is not a longitudinal study, but mainly a retrospective study based on the clinical observations of the patients included in the 2007 paper.

Present day observations of muscle weakness and if this had changed in the patients that were included in the 2007 paper would greatly add value to the present paper.

Author Response

Response to Reviewer 1 Comments

Point 1: The paper by Joshi et al presents data from a cohort of 60 patients with McArdles disesae (glycogen storage disease type V/myophosphorylase deficiency). The paper is well written and clearly presents data.

Our response: We are grateful to the reviewer for reviewing our manuscript and providing us with his/her constructive comments.

Point 2: However, there are some concerns about the paper. This present paper is based mainly on patients, whose genotype and clinical features were presented in 2007 by Deschauer et al. The new data is mainly histology and biochemistry and a revisit of the muscle weakness, also for a few added patients. This means that part of the data has already been published.

The presented histology and biochemical data does not contribute to the body of knowledge, since it has been known for a long time that in patients with McArdle disease lactate is low, ammonia is high due to alternate ATP production and periodic acid Schiff stain will demonstrate patches of glycogen in muscle sections.

Our response: We agree with the reviewer that the partial data (genetic and clinical aspects) of the patients has already been published. However, we have analysed other aspects in a large cohort (biochemical and histological as well as clinical data in more detail) that add to the knowledge of McArdle disease.

Point 3: In the original 2007 paper 15 of 40 patients had permanent weakness (37.5%), the present report states that 47% have permanent weakness. The meta-analysis demonstrates a 28% permanent weakness and a report from 2009 by Nadaj-Pakleza including 80 patients of which 11% had muscle weakness (rising to 37.5% if only counted above the age of 40). At this point it is generally accepted that there is no genotype-phenotype relationship, the 2007 paper states this as well. The muscle weakness generally affects arms and legs and it is possible that permanent weakness is related to age.

Our response: Our 2007 paper reported clinical data of 40 patients. Here, we have collected clinical data of 20 additional patients. Our analysis revealed permanent weakness in 29/60 (47%) patients. All patients with non-childhood onset had permanent muscle weakness suggesting age related permanent weakness as reported by Nadaj-Pakleza in 2009. This information is conveyed in discussion section (Line 178-183).

Point 4: It is of interest to the clinician how muscle weakness develops over time. However, it is my understanding that the presented clinical data (muscle weakness) is not a longitudinal study, but mainly a retrospective study based on the clinical observations of the patients included in the 2007 paper. Present day observations of muscle weakness and if this had changed in the patients that were included in the 2007 paper would greatly add value to the present paper.

Our response: The reviewer has raised a valid point: It would be really interesting to monitor course of muscle weakness over time. However, we were unable to include this information in our study. As the reviewer points out that we have performed the retrospective study, we do not have clinical information of patients over time.

Reviewer 2 Report

The authors present a large retrospective study on 60 Mcardle patients and describe the genotype of these patients along with several laboratory assessments of the tissues.  While the large amount of genotype data add to the current body of literature to other published studies, in general the study can be improved significantly in both methods and presentation in the manuscript.

Major concerns

The manuscript lacks clarity and is not linear in its progression. Specific needs follow:

The introduction lacks a well-defined purpose of the study. It does not clearly describe what is known in current literature and the gap in knowledge that this study will fill. Data is presented in the discussion (Figure 3)

line 172 many report misdiagnosed as psychosomatic disorders. Author state “many” etc. of patients report this. Authors should quantify these statements.

Results in figure 2 are presented with positive or negative controls. Although it is easy to believe that these patients had no myophosphorylase content or activity, without control groups/measurements it cannot be confirmed. Authors need to make define the positive control used in the methods section

Authors overstate data, jump to conclusions not supported by the data, or do not provide logical reasoning to statements.

Line 159 – data does not support the assumption/declaration that they should have been higher. Line 198- without controls, this cannot confirm data Line 222 – two patients does not support the close link between Lebanese and Turkish ethnicities. Provide supporting references or remove overstated data. Line 233 – need more that n=2 for milder symptoms. Permeant shoulder weakness does not add to this

Line 185- contradiction of statements. Authors state that under diagnosis/misdiagnosis occurs and then suggest that CK is primary criteria. This would lead to a large # of errors because of the numerous diseases that increase CK.

Table 4 is not reference in the results.

Minor comments:

Line 66 – further description of control subjects and measurements.

Line 69 = where were biopsies obtained?

Line 91 – define second wind phenomenon

Section 3.2  - better describe what CK is, why it increases with McArdles. Same with Ammonia.

Section 4.4 – restates the results.

Line 148 – define/contrast fixed weakness to permanent weakness

Author Response

Response to Reviewer 2 Comments

Point 1: The authors present a large retrospective study on 60 Mcardle patients and describe the genotype of these patients along with several laboratory assessments of the tissues.  While the large amount of genotype data add to the current body of literature to other published studies, in general the study can be improved significantly in both methods and presentation in the manuscript.

Our response: We thank the reviewer for going through our manuscript and for the valuable comments. As suggested by the reviewer, we have improved the methods and presentation in the manuscript. This was also suggested by other reviewers. The discussion section has also been improved for better interpretation of the results.

Major concerns

Point 2: The manuscript lacks clarity and is not linear in its progression.

Our response: We have worked on the manuscript to make it more linear in its progression.

Point 3: The introduction lacks a well-defined purpose of the study. It does not clearly describe what is known in current literature and the gap in knowledge that this study will fill.

Our response: The purpose of the study has now been included in introduction section (Line: 39-45)

Point 4: Data is presented in the discussion (Figure 3)

Our response: Figure 3 is only for reference purpose to convey that McArdle patients do not generally have marked muscle abnormalities.

Point 5: line 172 many report misdiagnosed as psychosomatic disorders. Author state “many” etc. of patients report this. Authors should quantify these statements.

Our response: ‘Many’ in line 172 has been specified (now line 197).

Point 6: Results in figure 2 are presented with positive or negative controls. Although it is easy to believe that these patients had no myophosphorylase content or activity, without control groups/measurements it cannot be confirmed. Authors need to make define the positive control used in the methods section.

Our response: Controls groups have been defined in methods section (line 84-85).

Point 7: Authors overstate data, jump to conclusions not supported by the data, or do not provide logical reasoning to statements.

Our response: This point has been addressed now and the text has been tailored to exclude the conclusions that are not supported by data.

Point 8: Line 159 – data does not support the assumption/declaration that they should have been higher.

Our response: This assumption has been deleted.

Point 9: Line 198- without controls, this cannot confirm data.

Our response: Line 198 reflects only the assumption that mutated alleles in McArdle disease are either not transcribed or not translated to protein.

Point 10: Line 222 – two patients does not support the close link between Lebanese and Turkish ethnicities. Provide supporting references or remove overstated data.

Our response: The overstated data have, now, been removed from the manuscript.

Point 10: Line 233 – need more that n=2 for milder symptoms. Permeant shoulder weakness does not add to this.

Our response: We agree with the reviewer that data of only 2 patients do not add to the conclusion. However, we only wanted to give a hint that splice site mutation might be more severe than other mutation. This message has been clearly conveyed in the manuscript.

Point 11: Line 185- contradiction of statements. Authors state that under diagnosis/misdiagnosis occurs and then suggest that CK is primary criteria. This would lead to a large # of errors because of the numerous diseases that increase CK.

Our response: Generally, McArdle patients are not very well diagnosed. Our message was that CK should be considered one of the criteria in diagnosis of McArdle patients, if the patients have clear clinical symptoms that are in line with McArdle disease. This has now been clarified in the manuscript (Line 209-210).

Point 12: Table 4 is not reference in the results.

Our response: Tables are renumbered and all tables are now referenced in results.

Minor comments:

Point 13: Line 66 – further description of control subjects and measurements.

Our response: Control subjects are now described.

Point 14: Line 69 = where were biopsies obtained?

Our response: The biopsies were obtained in our department. This information is added in the manuscript (Line: 76).

Point 15: Line 91 – define second wind phenomenon

Our response: Second wind phenomenon is now defined in the manuscript (Lines: 67 & 176).

Point 16: Section 3.2  - better describe what CK is, why it increases with McArdles. Same with Ammonia.

Our response: Elevated CK and mechanism of increased ammonia production in MCardle disease have been discussed in Discussion section (Section 5.2).

Point 17: Section 4.4 – restates the results.

Our response: Our results are correlated with results of previous studies in section 4.4.

Point 18: Line 148 – define/contrast fixed weakness to permanent weakness

Our response: The terms permanent weakness and fixed weakness were used as synonyms. Now only permanent weakness has been uniformly used in the manuscript for clarity.

Reviewer 3 Report

In this study, it was shown that 68% of the McArdle patients in the investigated population had a p.Arg50Ter mutation. Many had, in addition to this mutation, also other mutations. It appears that assessing p.Arg50Ter is the first to determine in the diagnosis of McArdle disease.

In the Introduction it is stated that the objective was to see ‘whether the clinical and biochemical phenotypes are influenced by the underlying genotypes’. This objective is not reflected by the abstract. Can you please amend the abstract to ensure that this objective and the results related to this objective are reflected in the abstract.

In the results you do not seem to address the objective. I had expected to see whether NH3 accumulation, lactate accumulation, glycogen storage etc differed between patients with different genotypes. That was what you said in the Introduction was your objective. So, can you relate clinical characteristics, measurements you di to genotype? Where some things more severe in a rare than in the p.Arg50Ter genotype for instance?

Line 227: The word’ evolutionary’ is superfluous. It just arose and has nothing to do with evolution.

Author Response

Response to Reviewer 3 Comments

Point 1: In this study, it was shown that 68% of the McArdle patients in the investigated population had a p.Arg50Ter mutation. Many had, in addition to this mutation, also other mutations. It appears that assessing p.Arg50Ter is the first to determine in the diagnosis of McArdle disease.

Our response: We thank the reviewer for his/her valuable comments in improving the quality of our manuscript.

Point 2: In the Introduction it is stated that the objective was to see ‘whether the clinical and biochemical phenotypes are influenced by the underlying genotypes’. This objective is not reflected by the abstract. Can you please amend the abstract to ensure that this objective and the results related to this objective are reflected in the abstract.

Our response: The abstract is now amended as suggested by the reviewer and the objective has been added there (Line: 10-11).

Point 3: In the results you do not seem to address the objective. I had expected to see whether NH3 accumulation, lactate accumulation, glycogen storage etc differed between patients with different genotypes. That was what you said in the Introduction was your objective. So, can you relate clinical characteristics, measurements you di to genotype? Where some things more severe in a rare than in the p.Arg50Ter genotype for instance?

Our response: Clear genotype-phenotype correlations based on clinical features, laboratory tests, biochemical and histological outcomes were not identified. This message is elaborated in the manuscript (Line: 260-261).

Point 4: Line 227: The word’ evolutionary’ is superfluous. It just arose and has nothing to do with evolution.

Our response: This message has been omitted from the manuscript.

Round 2

Reviewer 1 Report

The authors have addressed my comments satisfactorily.

Author Response

Comment of Reviewer #1:

Comment: The authors have addressed my comments satisfactorily.

Our Response: We are pleased that we were able to address the comments' of this Reviewer. We are obliged to the Reviewer for his/her comments during the first round of reviewing.

Reviewer 2 Report

The authors have done good job address the majority of the concerns. 

One very minor comment: The authors state the objective is to identify a possible genotype/phenotype. Often this means that the paper will run a statistical analysis for correlation among variables, however this was not conducted in the study. 

Author Response

Our Response to the comments of the reviewer:

Reviewer: The authors have done good job address the majority of the concerns. 

Our Response: We are pleased that the reviewer is satisfied with our corrections.

Reviewer: One very minor comment: The authors state the objective is to identify a possible genotype/phenotype. Often this means that the paper will run a statistical analysis for correlation among variables, however this was not conducted in the study.

Our Response: The statistical Analysis was performed to identify possible genotype-phenotype correlation. This has now been elaborated in methods section (lines 90-94) and has been discussed in discussion section (lines: 260-273).

Reviewer 3 Report

The changes made by the authors have considerably improved the manuscript. However, I recommend that the authors link back to the objective in the conclusion. At present the conclusion has nothing whatsoever to do with the objective.

Author Response

Our response to comments of Reviewer #2:

Comment: The changes made by the authors have considerably improved the manuscript. However, I recommend that the authors link back to the objective in the conclusion. At present the conclusion has nothing whatsoever to do with the objective.

Our response: We are pleased that the reviewer is almost satisfied with our revised manuscript. As the reviewer has suggested, we have modified the conclusion by linking it back to our objective. (Lines: 281-285). We thank the reviewer for his/her valuable time in going through our manuscript.